# The Prevalence of Immediate Hypersensitivity Reactions to the BNT162b2 mRNA Vaccine against SARS-CoV-2: Data from the Vaccination Campaign in a Large Academic Hospital

**DOI:** 10.3390/vaccines11050903

**Published:** 2023-04-27

**Authors:** Giovanni Paoletti, Jack Pepys, Maria Chiara Bragato, Sandro Paoletti, Alessandra Piona, Maria Rita Messina, Francesca Racca, Sebastian Ferri, Emanuele Nappi, Giovanni Costanzo, Lorenzo Del Moro, Francesca Puggioni, Giorgio Walter Canonica, Elena Azzolini, Enrico Heffler

**Affiliations:** 1Personalized Medicine, Asthma and Allergy, IRCCS Humanitas Research Hospital, Rozzano, 20089 Milan, Italy; giovanni.paoletti@hunimed.eu (G.P.); jackpepys@gmail.com (J.P.); mariachiara.bragato@st.hunimed.eu (M.C.B.); mariarita.messina@humanitas.it (M.R.M.); francesca.racca@humanitas.it (F.R.); sebastian.ferri@humanitas.it (S.F.); emanuele.nappi@humanitas.it (E.N.); giovanni.costanzo@humanitas.it (G.C.); lorenzo.delmoro@humanitas.it (L.D.M.); francesca.puggioni@humanitas.it (F.P.); giorgio_walter.canonica@hunimed.eu (G.W.C.); 2Department of Biomedical Sciences, Humanitas University, Pieve Emanuele, 20090 Milan, Italy; elena.azzolini@humanitas.it; 3Primary Care Pediatrics, ASL, 09121 Cagliari, Italy; dott.paoletti@tiscali.it; 4Allergy Service, Humanitas San Pio X Hospital, 20159 Milano, Italy; alessandra.piona@sanpiox.humanitas.it; 5Department of Clinical and Experimental Medicine, University of Florence, 50134 Florence, Italy; 6Clinical Quality Department, IRCCS Humanitas Research Hospital, Rozzano, 20089 Milan, Italy

**Keywords:** SARS-CoV-2, COVID-19, vaccine, allergy, hypersensitivity

## Abstract

The anti-SARS-CoV-2 vaccination has probably been the most effective tool for preventing the infection and negative outcomes of the COVID-19 disease, and therefore for interrupting the pandemic state. The first licensed SARS-CoV-2 vaccine was BNT162b2, an mRNA vaccine that has been widely used since the earliest stages of the global vaccination campaign. Since the beginning of the vaccination campaign, some cases of suspected allergic reactions to BNT162b2 have been described. Epidemiological data, however, have provided reassuring results of an extremely low prevalence of these hypersensitivity reactions to anti-SARS-CoV-2 vaccines. In this article, we describe the results of a survey carried out through the use of a questionnaire, administered to all the health personnel of our university hospital after the first two doses of the BNT162b2 vaccine, which investigated the development of adverse reactions after a vaccination. We analyzed the responses of 3112 subjects subjected to the first dose of the vaccine; among these, 1.8% developed symptoms compatible with allergic reactions and 0.9% with clinical manifestations of possible anaphylaxis. Only 10.3% of the subjects who had allergic reactions after the first injection experienced similar reactions after the second dose and none of them experienced anaphylaxis. In conclusion, the anti-SARS-CoV-2 vaccination is rarely associated with severe allergic reactions and the second dose of vaccine is safe for this group of patients.

## 1. Introduction

Amid the ongoing COVID-19 outbreak, vaccination has been widely hailed as the most effective means of mitigating the spread of the virus and preventing severe illness and death. This intervention stimulates the immune system to create defenses against the virus, which can effectively reduce the likelihood of its viral transmission and subsequent serious illness. As of now, more than 5.5 billion people have received the vaccine, making it an effective approach [1].

The vaccination triggers a strong immune response and has been discovered to be safe and effective among the majority of vaccinated individuals. However, as with all vaccines, there can be potential side effects. The most common symptoms include fatigue and soreness near the injection site, typically lasting only a few days, while serious adverse reactions are rare occurrences [2].

Hypersensitivity to vaccination and its excipients is a well-documented yet rare side effect of several vaccines [3]. 

The first vaccine against Severe Acute Respiratory Syndrome Coronavirus 2 (SARS-CoV-2) (the virus responsible for COVID-19) used on a large scale was BNT162b2 [4], which has been reported to be associated with common adverse effects including pain, swelling and redness at the injection site, fatigue, headache, muscle pain, chills, joint pain, fever, nausea, malaise, and lymphadenopathy [2]. Rarely have hypersensitivity reactions to BNT162b2, including delayed reactions suggesting cellular hypersensitivity, been reported [5]; the vaccine component that has been hypothesized to be involved as the potential eliciting allergen in patients with BNT162b2-induced hypersensitivity reactions is the excipient polyethylene glycol (PEG), a hydrophilic polymer that is frequently used as an excipient in everyday products including medicines, cosmetics, and foods [6,7,8]. As for hypersensitivity reactions to BNT162b2, although rare, there are some reported cases of both allergic reactions and anaphylaxis [9]. However, there are no large-scale studies investigating the extent of these hypersensitivity reactions to the vaccine.

The commonly recognized signs and symptoms of anaphylaxis are generalized urticaria, a diffuse erythematous rash, angioedema, respiratory and airway obstruction symptoms, nausea and other abdominal complaints, and hypotension. Albeit rarely, these symptoms have also been reported after the administration of anti-SARS-CoV-2 vaccines [10].

The objective of this paper is to describe the distribution of adverse events related to the BNT162b2 mRNA anti-SARS-CoV-2 vaccine in a large population of staff from an Italian Academic Hospital, specifically focusing on suspect hypersensitivity reactions that manifested on the day of the vaccine administration.

## 2. Materials and Methods

### 2.1. Study Design and Online Questionnaire

All healthcare workers and university staff from our center who received the first two doses of the BNT162b2 anti-SARS-CoV-2 vaccine 21 days apart were invited to fill in an online questionnaire (Appendix A) that was emailed to all the subjects following each vaccine dose. The questionnaire was created using a dynamic Google Form with a dropdown menu, including further questions based on the information provided. We asked all the participants to report their basic demographic data, including sex and age, and whether they experienced any adverse reaction following the vaccine administration. For those who reported a suspected adverse reaction, we asked them to specify their symptoms by selecting from a list of common adverse reactions. We also included an “other” option and requested a free-text elaboration from the participants who selected this option. We then asked for further information about the suspected adverse reaction, including its time of onset, its duration, and whether any drugs or an emergency department admission were required. Additionally, we inquired about relevant past medical history, including any known allergies, previous reactions to vaccinations, a history of COVID-19 infection, and any vaccination taken in the preceding four months. Further comments were possible throughout the questionnaire.

A database of the patients who reported adverse reactions after the vaccine administration has been collected. The reactions reported as occurring on the same day as the vaccine administration were considered to be “immediate reactions”. Among the reactions defined as “immediate”, those characterized by at least one clinical manifestation compatible with an immediate hypersensitivity reaction (a skin rash, urticaria, angioedema, sudden abdominal pain and/or nausea and/or diarrhea, symptoms compatible with bronchospasm, hypotension, and a loss of consciousness) were classified as “probable allergic reactions”. The latter were classified as possible/probable anaphylaxis, according to the Brighton Collaboration case definition of anaphylaxis, which includes dermatological, cardiovascular, respiratory, and gastrointestinal involvement [11].

The survey compiled after the second dose of the vaccine was evaluated to investigate whether the patients who had had an immediate adverse reaction, possibly of an allergic type, had undergone the second injection of the vaccine and had developed a similar reaction.

### 2.2. Brighton Collaboration Classification

The Brighton Collaboration case definition defines anaphylaxis as a clinical syndrome characterized by sudden onset, the rapid progression of its signs and symptoms, and the involvement of multiple (≥2) organ systems. In particular, the organ systems taken into consideration are the skin, respiratory system, cardiovascular system, and gastro-intestinal system [11].

Using this classification, anaphylaxis is classified based on its type of systemic manifestation, with several major and minor criteria. The major criteria include events such as generalized urticaria, generalized erythema, and upper airway swelling. The minor criteria comprise lighter symptoms, such as localized injection site urticaria, red and itchy eyes, and a persistent dry cough. 

After having classified these manifestations as major or minor, depending on the pattern manifested by the patient, there are three levels of certainty of an anaphylactic reaction, with 1 being the highest confidence level.

The patterns in the levels are divided as follows:

-Level 1: ≥1 major dermatological and ≥1 major cardiovascular and/or ≥1 major respiratory criterion.-Level 2: ≥1 major cardiovascular and ≥1 major respiratory criterion, or ≥1 major cardiovascular or respiratory criterion and ≥1 minor criterion involving ≥1 different system (other than cardiovascular or respiratory systems), or ≥1 major dermatologic and ≥1 minor cardiovascular and/or minor respiratory criterion.-Level 3: ≥1 minor cardiovascular or respiratory criterion and ≥1 minor criterion from each of ≥2 different systems/categories.

We used the criteria of the Brighton Collaboration case definition of anaphylaxis [11] with the aim of obtaining an estimation of the patients with a possible anaphylactic reaction in our cohort, assigning the Brighton Collaboration classification level of certainty for each reaction.

### 2.3. Statistical Analysis

The statistical analysis was performed using the SPSS 20.0 software (SPSS, Chicago, IL, USA). A Kolmogorov–Smirnov test was used to evaluate the normality of the distribution of each continuous variable, and depending on the result of this test, a Student t-test or Mann–Whitney test were used to compare the continuous variables. The categorical variables were compared with the Chi-square test. The continuous variables were presented as mean ± standard deviation (SD). *p* values of <0.05 were considered to be statistically significant.

## 3. Results

The questionnaire was sent to 4510 people and 4300 (95.3%) responses were obtained. Of which, 3112 (72.4%) were submitted following the first vaccination dose and 1188 (27.6%) following the second administration dose, but not after the first dose of the vaccine.

Of the responders following the first dose (n = 3112), 1212 (38.9%) were male and their average age was 40.3 ± 12.3 years. Atopic tendencies were declared by 346 (11.1%) responders, 175 (5.6%) reported a previous COVID-19 infection, and 993 (31.9%) referred a suspected adverse reaction, of which, 706 (71.1%) occurred on the day of the vaccination and were therefore classified as “immediate reactions”, according to the methods used in this study. In two cases (0.3% of those referring an immediate reaction), this adverse reaction resulted in an emergency room admission.

Immediate reactions were more frequent in females, atopics, and subjects who had contracted SARS-CoV-2 infection before the vaccination, and the subjects were younger than those who did not experience an immediate reaction (Table 1).

The most frequent adverse events occurring in the subjects who experienced immediate reactions were localized erythema at the injection site (643 subjects, 91.1%), asthenia (364 subjects, 51.6%), headache (252 subjects, 35.7%), myalgia (178 subjects, 25.2%), shivers (112 subjects, 15.9%), and arthralgia (110 subjects, 15.6%). Table 2 reports the reported adverse events and their relative frequency in the patients who experienced an immediate reaction after the first injection of the SARS-CoV-2 vaccine.

Fifty-seven subjects (8.1% of those who reported an immediate reaction and 1.8% of all survey responders after the first dose of the vaccine injection) had at least one clinical manifestation that was consistent with a probable allergic reaction. Among these, manifestations were more frequent in females and subjects with a previous SARS-CoV-2 infection than in subjects without a probable allergic reaction (Table 3).

According to the Brighton Collaboration classification, 27 patients (47.4% of those with probable allergic reactions and 0.9% of all survey responders) had probable anaphylaxis, with the following levels of probability: level 1 in 1 subject (1.8% of subjects with probable allergic reactions and 0.03% of all survey responders), level 2 in 20 subjects (35.1% of subjects with probable allergic reactions and 0.6% of all survey responders), and level 3 in 6 subjects (10.5% of subjects with probable allergic reactions and 0.2% of all survey responders). Only two patients with suspected allergic reactions reported being admitted to the emergency department following the reaction: the patient classified as level 1 and one patient classified as level 2 according to the Brighton Collaboration case definition.

Of the 3112 questionnaire respondents following the injection of the first dose of the SARS-CoV-2 vaccine, 1659 (53.3%) completed the survey after the second dose as well. Respondents who reported an immediate adverse reaction after the first vaccine injection were more likely to complete the second questionnaire as well (442 out of 706, 62.6% vs. 1217 out of 2406, 50.6%; *p* < 0.001). Thirty-nine of the subjects who had an immediate reaction to the first dose of the vaccine and also responded to the questionnaire after the second administration had a probable allergic reaction to the first dose; among these, only four (10.3%) reported allergic symptoms after the injection of the second dose of the vaccine, and in no cases were there clinical features consistent with anaphylaxis.

## 4. Discussion

For this study, a large number of vaccinated patients were taken into consideration, with more than 3000 respondents. The questionnaire’s goal was to estimate the number of people who may have had an allergic reaction after receiving a dose of vaccine. Based upon the answers obtained, any type of immediate adverse reaction after the first dose of the BNT162b2 vaccine accounted for about 23% of cases, but this was only 1.8% when restricting to symptoms that were compatible with a possible allergic reaction. This is interesting because, on the one hand, this is in line with what has already been reported on the clear prevalence of females among those who develop allergic reactions to anti-SARS-CoV-2 vaccines [10], and on the other, because it suggests that a previous infection with the virus can play the potential role of a primer for the development of hypersensitivity reactions to mRNA vaccines (as similarly described for more generically considered adverse events) [12]. Reinforcing this concept is the fact that previous SARS-CoV-2 infection was not significantly associated with the development of non-immediate and non-allergic reactions to vaccines.

Furthermore, only 0.9% of the survey responders experienced a systemic hypersensitivity reaction, with only one subject with clinical manifestations that could be classified as probable anaphylaxis according to the Brighton Collaboration classification, therefore determining the prevalence of probable anaphylaxis in around 3 cases out of 1000 individuals. Interestingly, among those who responded to both surveys (after the first and the second vaccine doses), only 10% of the subjects who experienced a suspect allergic reaction after the first dose experienced a new similar reaction after the second dose, and none of them reported an anaphylactic-like reaction.

Even though a large number of patients were asked to complete the questionnaire, this type of data analysis presented certain limitations, such as the subjectivity of the responses, which were not confirmed by a clinician. In fact, the data analyzed were not clinically validated, but rather self-reported by the responders, which rendered the responses of some subjects difficult to interpret, and in any case, frankly subjective. All of this is particularly true when we define “possible/probable anaphylaxis” as a multi-organ systemic allergic reaction that occurred on the same day as the vaccine being administered; however, since the clinical information was retrieved a posteriori and only through the completion of a closed-ended questionnaire, we believe that the definition that we chose is the most accurate possible, especially as it is corroborated by the use of the Brighton Collaboration case definition. Moreover, after the administration of the vaccine, the completion of the questionnaire was entirely voluntary; due to this mode of questionnaire administration, some subjects completed only one questionnaire and we do not know if the second dose was administered or not or why the second questionnaire was not completed.

Despite these limitations, the results of this study should be contextualized in the scientific literature published so far. In a systematic review and meta-analysis with more than 26 million mRNA SARS-CoV-2 vaccine recipients involved, the estimated prevalence of anaphylaxis after both vaccines was 5.58 per million doses administered (0.00058%), while the overall pooled prevalence estimate of non-anaphylactic reactions after both vaccines was 89.53 per million doses administered (0.008953%). This study analyzed 26 observational studies involving very large numbers of recipients, highlighting with its findings the exceedingly low incidence of anaphylaxis or other allergic reactions following a SARS-CoV-2 vaccination [13]. Our results show a much higher prevalence of probable anaphylaxis, but the nature of our study’s survey, which was based on the self-description of symptoms, certainly overestimates the prevalence of each adverse event compared to that which can be directly objected by healthcare personnel or retrospectively confirmed by a full clinical evaluation.

Furthermore, Robinson et al. examined the number of allergic reactions and anaphylaxis reported following a vaccination with SARS-CoV-2 mRNA. The following are their outcomes: of over 60 thousand employees who received their first dose of an anti-SARS-CoV-2 mRNA vaccine, symptom surveys were completed by over 50 thousand employees. Self-reported allergic symptoms were reported by 2.5% of these employees after the first dose. The patients with self-reported allergic symptoms after the first vaccine dose received the second dose in 71% of cases, with 17% of them reporting allergic symptoms again, but none of these were severe. Indeed, in this study, the vast majority of individuals (97%) with self-reported allergic symptoms safely completed the vaccination series [14]. Other studies have focused on patients who previously had a reaction to the first dose of the vaccine; in one study [15], of the 70 patients who received the second mRNA COVID-19 vaccine dose (88%), 62 had either no reaction or a mild reaction managed with antihistamines (89%) and only 2 patients required epinephrine treatment. In addition, in another study [16] with 189 patients, a total of 159 patients (84%) received a second dose. A prophylactic antihistamine administration prior to the administration of the second dose was given to 47 patients (30%). All 159 patients, including 19 individuals with first-dose anaphylaxis, tolerated the second dose. Our results are therefore in line with what has been described previously: only one out of ten patients who had experienced suspected allergic reactions after the first dose of the vaccine developed similar clinical manifestations after the second dose, and none of them experienced anaphylaxis. This phenomenon can be explained by the fact that the described reactions, although clinically compatible with type I immediate hypersensitivity reactions, can largely be explained by other more or less non-specific mechanisms of the direct activation of mast cells and/or basophils [17], which cannot be inevitably repeatable upon a subsequent administration of the vaccine. Another possible explanation, which does not have effective confirmation from the data retrieved from the questionnaire, is that the subjects who experienced immediate hypersensitivity reactions to the first dose of the vaccine carried out some premedication (for example with antihistamines), which may have reduced the likelihood of a recurrence of the allergic reaction to the second dose of the vaccine.

## 5. Conclusions

In conclusion, anti-SARS-CoV-2 vaccination is rarely associated with severe allergic reactions and anaphylaxis, as shown by the data presented in this article. In order to classify anaphylaxis, it is essential to use a model such as the Brighton Collaboration classification, as it is necessary to distinguish anaphylaxis from other symptoms such as vasovagal reactions and anxiety-related symptoms. The Brighton Collaboration classification aids in the identification and subsequent treatment of anaphylaxis by preventing the mislabeling of other conditions, especially in the absence of objective findings.

In addition, these results indicate that the second dose of the vaccine is safe for this group of patients, even if there were allergic reactions following the initial dose.

## Figures and Tables

**Table 1 vaccines-11-00903-t001:** Main characteristics of subjects experiencing immediate vs. non-immediate reactions after the first administration of BNT162b2 mRNA anti-SARS-CoV-2 vaccine.

	Immediate Reactions (n = 706)	Non Immediate Reactions (n = 2406)	*p* Value
Females, n (%)	527 (74.6%)	1373 (57.1%)	*p* < 0.001
Mean age ± SD	37.5 ± 11.1	40.9 ± 13.4	*p* < 0.001
Atopics, n (%)	245 (34.7%)	101 (4.2%)	*p* < 0.001
Previous SARS-CoV-2 infection, n (%)	106 (15.0%)	69 (2.9%)	*p* < 0.001

**Table 2 vaccines-11-00903-t002:** Adverse events and their frequency in patients experiencing an immediate reaction after SARS-CoV-2 vaccine first injection.

Adverse Event		Relative Frequency, n (%)	
Reaction at injection site		643 (91.1%)	
Asthenia		364 (51.6%)	
Headache		252 (35.7%)	
Myalgia		178 (25.2%)	
Shivers		112 (15.9%)	
Arthralgia		110 (15.6%)	
Fever		62 (8.8%)	
	<38 °C		52 (7.4%)
	≥38 °C		10 (1.4%)
Nausea		55 (7.8%)	
Sweating		44 (6.2%)	
Insomnia		37 (5.2%)	
Lightheadedness		31 (4.4%)	
Tachycardia		29 (4.1%)	
Lymphadenopathy		26 (3.4%)	
Localized skin rash		26 (3.4%)	
Diarrhea		23 (3.4%)	
Abdominal pain		19 (2.7%)	
Anxiety/panic attack		13 (1.8%)	
Dyspnea		12 (1.7%)	
Chest pain		10 (1.4%)	
Hypotension		7 (1.0%)	
Vomiting		6 (0.8%)	
Transient face paralysis		5 (0.7%)	
Diffused skin rash		4 (0.6%)	
Face edema		2 (0.3%)	
Tongue edema		2 (0.3%)	
Loss of consciousness		0 (0%)	

**Table 3 vaccines-11-00903-t003:** Main characteristics of subjects experiencing probable allergic reactions after the first administration of BNT162b2 mRNA anti-SARS-CoV-2 vaccine.

	Probable Allergic Reaction (n = 57)	Non Probable Allergic Reaction (n = 649)	*p* Value
Females, n (%)	50 (87.7%)	477 (73.5%)	0.018
Mean age ± SD	39.2 ± 11.2	37.4 ± 11.0	0.246
Atopics, n (%)	22 (38.6%)	223 (34.4%)	0.519
Previous SARS-CoV-2 infection, n (%)	14 (24.6%)	92 (14.2%)	0.036

## Data Availability

Not applicable.

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
