# Peer review of "The Prevalence of Immediate Hypersensitivity Reactions to the BNT162b2 mRNA Vaccine against SARS-CoV-2: Data from the Vaccination Campaign in a Large Academic Hospital"

_vaccines, 2023, doi:10.3390/vaccines11050903_

Round 1

Reviewer 1 Report

I was invited to revise the paper entitled "Prevalence of immediate hypersensitivity reactions to BNT162b2 mRNA vaccine against SARS-CoV-2: data from the vaccination campaign in a large Academic Hospital". It was a brief report aimed to evaluate the prevalence of adverse events related to
Pfizer anti-SARS-CoV-2 vaccine in a large population of an Italian  67
Hospital occurred on the day of vaccine administration.

All participants (healthcare workers) filled an online questionnaire about reations occurred after the vaccination administration. Possible anaphylaxis was classified according to Brighton Collaboration case definition.

The topic is relevant for public health and it can improve the knowledge on this field.

Major observation:

- Authors should evaluate the incidence of adverse event among patients priorly infected with Sars-cov-2.

Minor observations:

- Lines 136: Authors stated "Four thousand, six-hundred and sixty" but they reported in number 4310;

- In table 1, Authors should specify the tests performed;

- Statistical analysis section reported only Fisher's exact test for categorical variables but in table 1 Chi-squared seems to be performed;

Author Response

We sincerely thank the Reviewer for his/her appreciation, comments and suggestions.

Here you can you find the point-by-point response to your criticisms:

Major observation:

  • Authors should evaluate the incidence of adverse event among patients priorly infected with Sars-cov-2.

RESPONSE: As reported in the Results, Table 1 and Table 3, immediate adverse reactions and those events that were classifiable as probable allergic reactions were more frequent in patients with previous SARS-CoV-2 infection. As far as the non immediate and non allergic reactions, no difference was found between those who were priorly infected compared to those that did not experiences SARS-CoV-2 infection (we added this information in the revised Discussion)

Minor observations:

  • Lines 136: Authors stated "Four thousand, six-hundred and sixty" but they reported in number 4310;

RESPONSE: thanks...it was a typo! We corrected in the revised text (4300 responses in total).

  • In table 1, Authors should specify the tests performed;

RESPONSE: no tests was performed, as Table 1 reports the results of statistical analysis among responses to the online questionnaire.

  • Statistical analysis section reported only Fisher's exact test for categorical variables but in table 1 Chi-squared seems to be performed;

RESPONSE: Thanks...actually we used Chi-squared test. We corrected the revised Statistical methods.

Reviewer 2 Report

The manuscript of Poletti and collaborators describes the results of an online survey administered to the personnel of the hospital after  the first and second vaccination dose to investigate the adverse reactions.

The manuscript is well written and, even if it presents some limitations (described by the Authors), the results obtained are in line with those reported by other studies.

Minor revisions

- the questionnaire has to be published as supplementary material

-the initial number of questionnaire sent to paticipants should be indicated

-the authors declare 4310 responses: 3112 responses after the first dose and 1188 after the second dose, but the total is 4300 and not 4310. Could you explain?

- all the partecipants received the questionnaire after the first and also after the second dose?  Thus of 3112 responses after the first dose, 1659 completed the survey also after the second dose. And 1188 were those who responded only after the second dose but not after the first one? Please explain well these numbers

 - line 212: the results of this study the results of our study" please correct

Author Response

We sincerely thank the Reviewer for his/her appreciation, comments and suggestions.

Here you can you find the point-by-point response to your criticisms:

Minor revisions

  • the questionnaire has to be published as supplementary material

RESPONSE: an english language translation of the questionnaire has been submitted as supplementary material of the revised manuscript.

-the initial number of questionnaire sent to paticipants should be indicated

RESPONSE: we added this information in the Results

-the authors declare 4310 responses: 3112 responses after the first dose and 1188 after the second dose, but the total is 4300 and not 4310. Could you explain?

RESPONSE: thanks! it was a typo...we corrected in the revised Results (4300 is the correct number)

  • all the partecipants received the questionnaire after the first and also after the second dose?  Thus of 3112 responses after the first dose, 1659 completed the survey also after the second dose. And 1188 were those who responded only after the second dose but not after the first one? Please explain well these numbers

RESPONSE: 1188 were actually responding only to the second dose questionnaire, while 1659 were those who completed the second dose questionnaire among those who also fulfilled the first dose questionnaire (we added this clarification in the Results)

 - line 212: the results of this study the results of our study" please correct

RESPONSE: thanks...we corrected the sentence

Reviewer 3 Report

This manuscript reports on the incidence of adverse reactions to SASRS-CoV2 vaccination in a population of health workers. The authors broadly use the term “anaphylaxis” although there is little evidence that events occurring on the same day as the injection are indeed basophil/mast cell mediated. There should be some discussion about two major points:

-         - Specific IgE-mediated reactions or direct activation of basophils/mast cells

-         - Any idea about a previous sensitization, such immediate manifestations, when truly “allergic”, depending on a previous asymptomatic exposure.

There should be some discussion also about the low rate of clinical manifestations after the second vaccine, which argues against type I hypersensitivity, unless premedicaton (i.e. anti-histaminics) was applied, which should be detailed.

The possibility (and obviously not encountered) of delayed cellular hypersensitivity should be at least evoked.

Finally, more information about what was undertaken for the exploration of the most severe cases (i.e. ICU transfer) should be provided.

The manuscript is globally correctly written but some grammatical errors require attention. The style of the reference list should be harmonized.

Author Response

We sincerely thank the Reviewer for his/her appreciation, comments and suggestions.

Here you can you find the point-by-point response to your criticisms:

  • The authors broadly use the term “anaphylaxis” although there is little evidence that events occurring on the same day as the injection are indeed basophil/mast cell mediated.

RESPONSE: We agree with the Reviewer that the given definition of "possible/probable anaphylaxis" (see the Methods) is not exactly that of the international recommendations (e.g.: WAO Position Paper on anaphylaxis); however, being the clinical information retrieved a posteriori and only through filling out a closed-ended questionnaire, we believe that the definition we have chosen, especially as it is supported by the use of the Brighton Collaboration case definition, is as accurate as possible. In the revised manuscript we have reported a comment on this in the Discussions.

  • There should be some discussion about two major points:

         - Specific IgE-mediated reactions or direct activation of basophils/mast cells

RESPONSE: We thank the Reviewer for this comment. We have added some considerations about it in the Discussion

         - Any idea about a previous sensitization, such immediate manifestations, when truly “allergic”, depending on a previous asymptomatic exposure.

RESPONSE: Unfortunately, due to the specific nature of the study (derived solely from answers provided to a closed-ended questionnaire generically on adverse events occurring after vaccination) we do not have sufficient data to be able to hypothesize what possible allergenic sources may have induced primary sensitization in patients with true immediate type I reactions

  • There should be some discussion also about the low rate of clinical manifestations after the second vaccine, which argues against type I hypersensitivity, unless premedicaton (i.e. anti-histaminics) was applied, which should be detailed.

RESPONSE: We thank the Reviewer for this comment. We have added some considerations about it in the Discussion

  • The possibility (and obviously not encountered) of delayed cellular hypersensitivity should be at least evoked.

RESPONSE: We mentioned the delayed cellular hypersensitivity reactions in the revised Introduction.

  • Finally, more information about what was undertaken for the exploration of the most severe cases (i.e. ICU transfer) should be provided.

RESPONSE: The questionnaire did not include questions relating to which therapies were administered after the possible adverse reaction, while the data relating to the possible hospitalization in the emergency room is available which, in our series, was reported only in two cases (the case in which level 1 according to the Brighton Collaboration case definition, and one case with level 2 of the same classification). This information was reported in the Results of the revised manuscript

Round 2

Reviewer 1 Report

Authors properly addressed all comments. 

Author Response

Thank you very much!

Reviewer 3 Report

The authors have satisfactorily amended the manuscript. The sentence added as per request of the other reviewer about the number of questionnaires received could be clarified (not clear who "the latter" are)

Author Response

We corrected and clarified that "The questionnaire was sent to 4510 people, and 4300 (95.3%) responses were obtained. Of which 3112 (72.4%) were submitted following the first vaccination dose, and 1188 (27.6%) following the second administration but not after the first dose of vaccine."